# Exocrine and Endocrine Insufficiency in Autoimmune Pancreatitis: A Matter of Treatment or Time?

**DOI:** 10.3390/jcm11133724

**Published:** 2022-06-28

**Authors:** Sara Nikolic, Patrick Maisonneuve, Ingrid Dahlman, J.-Matthias Löhr, Miroslav Vujasinovic

**Affiliations:** 1Department of Medicine, Huddinge, Karolinska Institute, 14186 Stockholm, Sweden; sara.nikolic@ki.se (S.N.); ingrid.dahlman@ki.se (I.D.); 2Department of Gastroenterology, Clinic of Internal Medicine, University Medical Centre Maribor, 2000 Maribor, Slovenia; 3Division of Epidemiology and Biostatistics, IEO—European Institute of Oncology IRCCS, 20141 Milan, Italy; patrick.maisonneuve@ieo.it; 4Department of Upper Abdominal Diseases, Karolinska University Hospital, 14186 Stockholm, Sweden; matthias.lohr@ki.se; 5Department of Clinical Science, Intervention, and Technology (CLINTEC), Karolinska Institute, 14186 Stockholm, Sweden

**Keywords:** autoimmune pancreatitis, pancreatic exocrine insufficiency, diabetes mellitus, treatment

## Abstract

Background: Autoimmune pancreatitis (AIP) is a specific form of chronic pancreatitis with a high relapse rate after treatment. AIP patients are burdened with an increased risk of long-term sequelae such as exocrine and endocrine insufficiency. Our objective was to investigate if pharmacological treatment affects both endocrine and exocrine pancreatic function in patients with AIP. Methods: We included 59 patients with definite AIP in the final analysis. Screening for diabetes mellitus (DM) and pancreatic exocrine insufficiency (PEI) was performed at the time of AIP diagnosis and during follow-up. Results: There were 40 (67.8%) males and 19 (32.2%) females; median age at diagnosis was 65 years. Median follow-up after the diagnosis of AIP was 62 months. PEI prevalence at diagnosis was 72.7% and was 63.5% at follow-up. The cumulative incidence of DM was 17.9%, with a prevalence of DM at diagnosis of 32.8%. No strong association was found between pharmacological treatment and occurrence of PEI and DM. Univariate analysis identified potential risk factors for PEI (other organ involvement and biliary stenting) and for DM (overweight, blue-collar profession, smoking, weight loss or obstructive jaundice as presenting symptoms, imaging showing diffuse pancreatic enlargement, smoking). In a multivariate analysis, only obstructive jaundice was identified as a risk factor for DM both at diagnosis and during follow-up. Conclusions: Our results suggest that the prevalence of endocrine and exocrine insufficiency in AIP is high at diagnosis with an additional risk of PEI and DM during follow-up despite pharmacological treatment.

## 1. Introduction

Autoimmune pancreatitis (AIP) is a unique form of pancreatitis with two subtypes, AIP type 1 and AIP type 2, which are distinct in terms of scope and histology. AIP type 1 is a pancreatic manifestation of a systemic IgG4 related disease, histologically identified as lymphoplasmacytic sclerosing pancreatitis. On the other hand, AIP type 2 is not related to IgG4 and is histologically described as an idiopathic duct-centric pancreatitis. Diagnosis is not straightforward, and it is based on International Consensus Diagnostic Criteria (ICDC) [1]. Indications for treatment are symptomatic patients with clinical, laboratory and/or radiological signs of pancreatic involvement or other organ involvement (OOI). We also treat asymptomatic patients with either persistent pancreatic mass or persistent laboratory test abnormalities indicating active OOI [2]. The goal of treatment is to alleviate the symptoms and prevent further pancreatic damage and consequent complications. Steroids are the most successful first-line treatment for inducing remission; however, relapses are frequent—up to 55% [2]. It was hypothesised that steroids reverse the inflammation and the damage of pancreatic tissue [3,4,5], leading to recovery from exocrine and endocrine dysfunction, at least in the short term [6,7]. However, long-term sequelae, such as pancreatic exocrine insufficiency (PEI) and diabetes mellitus (DM) are often described at onset and during follow-up in both types of AIP [8].

The prevalence of DM among AIP patients in European studies ranges from 12% to 55% [9,10,11,12], whereas in Japanese studies, the prevalence ranges from 42% to 78% [13,14,15]. In most patients, DM is diagnosed simultaneously with the onset of AIP, but there are subgroups of patients diagnosed with DM before the onset of AIP or who develop DM as a complication of glucocorticoid (GC) treatment. [13,14,16]. PEI has been reported in 83%–88% of AIP cases [11,13]. There is no firm evidence regarding the implications of different types of pharmacological treatment for DM and PEI in patients with AIP, and the available data are controversial.

The aim of our study was to investigate the association between pharmacological treatment and both endocrine and exocrine pancreatic function in patients with AIP as well as to determine risk factors for PEI and DM development. 

## 2. Patients and Methods

We retrospectively analysed medical records of patients diagnosed with AIP at The Department of Upper Abdominal Diseases at Karolinska University Hospital, Stockholm, Sweden, between January 2001 and October 2020.

### 2.1. Inclusion Criteria

We included patents diagnosed with definite AIP according to the ICDC [1]. For patients diagnosed before publication of the ICDC (2011), two independent senior investigators retrospectively reviewed diagnoses and applied these criteria.

### 2.2. Exclusion Criteria

Patients were excluded if they were younger than 18 years at the time of analysis and had missing data from medical charts and follow-up from less than 12 months. We also excluded patients with AIP type 2, AIP not otherwise specified (NOS), probable AIP according to ICDC, AIP patients in whom diagnosis was confirmed after pancreatic surgery, and patients with a history of any major abdominal surgery due to the potential impact on false low values of faecal elastase-1 (FE-1) or postcibal asynchrony causing PEI [17].

### 2.3. Outcomes: Diagnosis of PEI and DM

Diagnosis of PEI was based on values of FE-1 expressed in μg/g of stools. Levels < 200 μg/g were categorised as mild PEI and levels < 100 μg/g as severe PEI [6]. Patients were routinely asked to categorise faeces according to the Bristol stool scale [18] and only faeces type 1–5 was used for FE-1 analysis (in order to avoid false negative values of FE-1 in liquid stools) [19]. DM was diagnosed by either fasting plasma glucose levels ≥ 7.0 mmol/L, plasma glucose ≥ 11.1 mmol/L two hours after a 75 g oral glucose load, casual plasma glucose ≥ 11.1 mmol/L or glycated haemoglobin (HbA1c) ≥ 6.5% (48 mmol/mol) [20]. A DM diagnosis required two blood samples that passed above these thresholds. FE-1, HbA1c and plasma glucose were measured several times during follow-up; however, to determine the presence of PEI and DM, we used values at diagnosis (before therapy) and last contact (after therapy).

### 2.4. Variables

Medical records were thoroughly analysed and the following data were collected: gender, age at AIP diagnosis, follow-up time in months (defined as the period between date of AIP diagnosis and the last contact with the patients), type of AIP diagnosis (type 1, type 2), past pancreatic and/or abdominal surgery, treatment of AIP, OOI, AIP recurrence after initial treatment, smoking status (defined as never, former and active smokers), alcohol intake (alcohol overconsumption was defined as >5 units per day), profession of patients (defined as white or blue collar profession), body mass index (BMI) which was categorized as underweight (<18.5), normal (18.5 ≤ 25), overweight (25.0 to 29.9) or obese (≥30.0), symptoms at diagnosis of AIP (abdominal pain, involuntary weight loss, obstructive jaundice, acute pancreatitis, new-onset diabetes), and IgG serology panel.

### 2.5. Treatment of AIP

Patients were categorized according to medical treatment with one or more of the following: GCs, azathioprine, biologics, and rituximab. Patients who received only biliary stenting were considered to have no medical therapy. GCs were indicated as first line AIP-related treatment [2]. Treatment with biologics was introduced by a rheumatologist or inflammatory bowel disease (IBD) specialist due to concurrent IBD or rheumatological disease, never due to AIP. Those agents were tumour necrosis factor (TNF) alfa inhibitors (adalimumab, infliximab, golimumab), interleukin 12 (IL-12) and interleukin (IL-23) inhibitor (ustekinumab), and integrin α_4_β_7_ inhibitor (vedolizumab). Consequently, we decided to analyse rituximab (a chimeric monoclonal antibody targeted against CD20 which is a surface antigen present on B cells) [21] separately, as it is relevant and recommended for AIP treatment [2].

### 2.6. Statistical Analysis

Data are expressed as a median, and interquartile range (IQR) for numerical data, or as a percentage for categorical data. A comparison of data was undertaken using appropriate non-parametric statistical tests: for categorical data chi-square or Fisher’s exact test, and for numerical data Mann–Whitney U test. We used the Kaplan–Meier method to estimate the cumulative incidence of DM during follow-up. For the sake of transparency regarding missing values, the total number of values was reported for each variable in Table 1. Later, percentages were calculated excluding missing values. The log-rank test was used to compare differences of cumulative incidence between groups. Analyses were performed using SAS software (version 9.4, SAS Institute, Cary, NC, USA); *p*-values < 0.05 (two-sided) were considered statistically significant.

### 2.7. Ethics

The study adheres to the latest version of the Declaration of Helsinki and was approved by the Swedish Ethical Committee (Etikprövningsmyndigheten Dnr. 2016/1571-31 and 2020-02209).

## 3. Results

A total of 159 patients with AIP were analysed (Figure 1). The final analysis, after implementation of inclusion and exclusion criteria, included 59 patients with definite AIP. There were 19 (32.2%) females; median age at diagnosis was 65 years (IQR = 49–71). The median follow-up after the diagnosis of AIP was 62 months (IQR 38–104). The main demographic, clinical, radiological, and treatment-related characteristics of the AIP type 1 patients in this study are presented in Table 1. 

### 3.1. PEI and AIP

At the time of diagnosis, before commencement of any treatment, 32/44 (72.7%) patients had PEI: 9 (20.5%) mild PEI and 23 (52.3%) severe PEI. During the follow-up, 3 additional patients developed PEI, and in 4 patients, pancreatic exocrine function normalized. Change of PEI class from severe to mild and vice versa occurred in 5 and 4 patients, respectively. Prevalence of PEI at the last follow-up was 33/52 (63.5%). Out of all exposures, smoking was inversely associated with PEI status at last contact, without any association at baseline. Biliary stenting was associated with a higher prevalence of PEI at follow-up (19/23 (82.6%) vs. 14/29 (48.3%), *p* = 0.02). On the other hand, OOI was associated with a higher prevalence of PEI both at diagnosis 32/41 (78.0%) vs. 0/3 (0%), *p* = 0.01 and last follow-up 33/59 (67.3%) vs. 0/3 (0%), *p* = 0.04 (Appendix A).

### 3.2. DM and AIP

At the time of AIP diagnosis, new-onset DM was diagnosed in 14/58 (24.1%) patients. Five patients (8.6%) had DM type 2 diagnosed before AIP diagnosis. Univariate analysis showed that being overweight, from blue-collar professions, a smoker, experiencing weight loss or obstructive jaundice as presenting symptoms, having diffuse pancreatic enlargement on imaging, and the necessity for biliary stenting were associated with the presence of DM at baseline. During the observation period, blue-collar professions, smoking, weight loss, obstructive jaundice, PEI at diagnosis, diffuse pancreatic enlargement on imaging, and stenting were associated with DM (Table 2). At multivariable analysis, smoking (OR = 4.92; 95% CI 1.22–19.9) and obstructive jaundice (OR = 9.47; 95% CI 2.06–43.5) were strongly associated with the presence of DM at diagnosis of AIP (Table 2). After an adjustment for smoking, the association between blue-collar professions and DM at diagnosis lost statistical significance. At multivariable analysis, obstructive jaundice was also significantly associated with the presence or development of DM during follow-up (HR = 4.92; 95% CI 1.75–13.9), while the association with smoking and blue-collar professions remain only of borderline significance.

A total of 7 (17.9%) of 38 patients with normal glucose values at diagnosis developed DM during follow-up. The Kaplan–Meier curve in Figure 2 shows the cumulative incidence of DM during follow-up, stratified by obstructive jaundice, smoking, and blue-collar professions at diagnosis. After 10 years of follow-up, 58.3% of all patients (95% CI 32.1–77.4) developed DM. IgG serology was not associated with DM cumulative incidence nor PEI and DM prevalence at diagnosis or last contact. 

### 3.3. Pharmacological Treatment, PEI, and DM

There were 51 (86.4%) patients who received pharmacological treatment and 8 (13.6%) patients without pharmacological treatment. Among those, 6 patients (10.2%) were without any treatment (patients with spontaneous regression of AIP). First-line treatment (GC) was given to 49 (83.1%) patients. Azathioprine was prescribed for 9 (15.3%) patients, rituximab for 10 (16.9%) patients, and biologics were prescribed by other specialists (IBD, rheumatologist) to treat 2 (3.4%) patients. No significant association was found between pharmacological treatment and occurrence of PEI and DM (there was only a slight tendency towards an association between rituximab treatment and PEI prevalence at last contact 9/10 (90.0%) vs. 24/42 (57.1%), *p* = 0.07). Interestingly, maintenance treatment was weakly associated with a higher prevalence of PEI at last contact: 17/21 (81.0%) vs. 16 (51.6%), *p* = 0.04. Detailed treatment-related results are shown in Appendix A.

## 4. Discussion

The aim of our research was to examine the loss of pancreatic exocrine and endocrine function in AIP patients and its relationship with pharmacological treatment. Our study of 59 definite AIP patients showed that during a median follow-up of 62 months, PEI prevalence in our cohort did not change significantly. The cumulative incidence of DM was 17.9%, and in all patients, DM was persistent. On the other hand, there were some fluctuations in PEI occurrence (Table 1).

Pharmacological treatment was not strongly associated with the occurrence of PEI or DM in our cohort. However, maintenance treatment and treatment with rituximab were weakly associated with a higher prevalence of PEI at last contact. However, those associations might be logical consequences of higher disease activity, as a more aggressive disease demands maintenance GC treatment or/and treatment with rituximab [2], suggesting that both maintenance treatment and rituximab were mediators between disease activity and PEI, without its own impact on PEI.

With aging, transformation of pancreatic volume, structure, and perfusion occur, which may impact exocrine function [22]. In our cohort, risk of PEI at baseline was not associated with older age, which is well established in the literature, possibly due to the median age of our patients, which was 65 years. Other organ involvement was associated with higher PEI occurrence both at diagnosis and at last contact. Additionally, risk of PEI during follow-up was weakly associated with biliary stenting, which strongly suggests there is a role of other organ involvement and PEI (biliary stenting was performed in patients with immune-related cholangitis, that has most common OOI in patients with autoimmune pancreatitis type 1). Univariate analysis identified several risk factors associated with DM at diagnosis and during follow-up (Table 2). Multivariable analysis showed that smoking, blue-collar professions, and obstructive jaundice were associated with the presence of DM at diagnosis. However, during follow up, only obstructive jaundice was strongly associated with DM with HR 4.92 (1.75–13.9), whereas the association with smoking and blue-collar professions remain only of borderline significance. Interestingly, 15 (25.4%) of our cohort were patients with blue-collar professions, which is in contrast with the findings from Maillette de Buy Wenniger et al. where both Amsterdam and Oxford cohorts of IgG4 cholangitis showed a higher prevalence of blue-collar professions, which were 88% and 61%, respectively [23].

So far, only a few studies (Table 3) with small sample sizes have focused on DM and PEI prevalence in AIP before and after treatment. In an Italian study of PEI and pharmacological treatment in 21 patients with AIP, the mean FE-1 values improved in all patients after GC treatment [6]. However, the duration of the disease was not stated, and in almost half of the patients, the category of PEI did not change; thus, the improvement could not be considered significant. Additionally, a Japanese study of just six patients showed that pancreatic function improved in 50% of the patients after GC treatment [24]. The rationale for the hypothesis that steroids might ameliorate or even reverse the loss of pancreatic function (“restitutio ad integrum”) lies behind the findings that pancreatic function is differently impaired in AIP compared with chronic pancreatitis (CP). Ito et al. found that, due to ductal stenosis, AIP patients have a lower volume of duodenal aspirate with normal bicarbonate and lower amylase concentrations in comparison with CP patients [25]. A further histological analysis of pancreatic tissue revealed that intact, but stenosed ductal basement membranes in AIP patients, compared with injured basement membranes in CP patients, were responsible for that difference [25]. This study also did not clearly state the duration of the disease in AIP and CP patients.

When it comes to diabetes, European guidelines for IgG4 digestive diseases state that GC treatment results in favourable consequences regarding DM in patients with AIP type 1, especially if there is a simultaneous onset of DM [2]. Studies from Japan described short-term improvement of 55–66% for simultaneous onset DM and up to 36–54% for pre-existing DM (worsening was reported in 9–15%). The improvement rate was either not defined at all or defined as a HbA1c decrease by more than 0.5%, plus a decreased dose of insulin for insulin users [7,14]. An analysis of the effects of long-term steroid treatment (up to 3 years) on DM in patients with AIP showed that DM improvement had risen to 63% with no reported worsening [7]. However, in comparison to our study, the above-mentioned Japanese studies concentrated on improved control and not the reversibility of DM. Additionally, a study by Noguchi et al. showed that after GC treatment, the prevalence of both pre-existing and concomitant DM decreased by 9% and 23%, respectively [26]. On the other hand, Masuda et al. concluded that pancreatic atrophy after treatment with GCs could be closely associated with the onset of DM in AIP patients [27], which was also Noguchi’s observation, as well as the fact that DM in patients with AIP was associated with impaired insulin secretion rather than insulin resistance in the early phase of AIP [26]. Similarly, Ito et al. observed a reduction in both β-cell (insulin) and α-cell (glucagon) secretion in patients with AIP [25], in contrast with DM type 2 where glucagon secretion was elevated [28]. Histological analysis revealed that islet cells of AIP patients were either intact or destroyed due to edema and ischemia because of fibrosis and lymphoplasmatic cell infiltration [25]. Thus, pancreatic atrophy might, at least partially, account for the irreversibility of DM in patients with AIP.

There are several limitations of our study. Despite having one of the largest AIP cohorts in Europe (which represents a strength of the study), statistical power is low due to a small sample size, which affects the precision of the study. Due to missing data regarding FE-1, mostly at diagnosis, we were unable to perform a longitudinal analysis of PEI; thus, the results regarding PEI must be interpreted with caution. However, the study results are still generalizable to other AIP cohorts. Despite the observational design, we restricted inclusion to only definite AIP patients and performed regression analysis with the intention to minimize confounding. We avoided comparing patients with and without pharmacological treatment due to possible bias by treatment indication. A detailed description of pancreatic (dys)function in patients with definite AIP type 1 was provided based on long-term follow-up, which represents the strength of this study. We hope that prescribed peroral medication was taken by our patients. If not, there could be a non-differential misclassification of exposure (pharmacological treatment) which would lead to the underestimation of the associations (dissolved association). The previously mentioned bias by misclassification of both exposure and outcome may be important in an analysis where no association was found. Future prospective studies are needed to assess the influence of pharmacological treatment on pancreatic exocrine and endocrine function; however, currently there is no evidence to assume steroids alone or combined with other agents reverse the loss of pancreatic function. Whether loss of pancreatic function is a matter of time or disease activity in the individual is yet to be studied.
jcm-11-03724-t003_Table 3Table 3Occurrence of PEI and DM in patients with AIP in different studies.Author, Year, CountryPatients (N)Method of PEI DiagnosisOccurrence of PEIOccurrence of DMFrulloni, 2010, Italy [6]21FE-1At AIP diagnosis 62% had severe PEI and 19% mild PEI.After CST, FE-1 levels increased in all patients.Within 1 month after CST, 33% continued to show severe PEI.Before CST, DM was diagnosed in 5 patients (24%), which increased to 10 patients (48%) during CST.The dosage of insulin was decreased after tapering of steroids and only 4 patients (19%) continued to require low-dose insulin therapy at the end of CST.Nishino et al., Japan, 2006 [15]12BT-PABABefore CST 6 (67%) of the 9 patients had reduced pancreatic exocrine function.After CST pancreatic exocrine function improved in 3 patients.10 patients (83.3%) had DM before CST, and in 3 patients HbA1c level improved after the CST. Two patients experienced a transient loss of glycemic control after CST.Nishimori, 2006, Japan [14]167-Not determined.66.5% of patients had DM.In the early-onset group 36% showed improvement of DM control, 45% showed no change, and 18% worsening.In the simultaneous-onset group 55% showed improvement of DM control, 29% showed no change, and 16% worsening.Miyazawa, 2017, Japan [16]82-Not determined.61.7% of patients had DM.37.5% showed improvement, 21.9% showed exacerbation, and 40.6% showed no change.Miyamoto, 2012, Japan [7]69BT-PABAPEI was reduced in 91% of AIP patients with DM. In all patients whose glucose tolerance improved after CST, pancreatic exocrine function also improved.46% had DM.Three months after starting CST, DM improved in 54% patients.At about 3 years after starting CST, DM improved in 63% of patients.Kamisawa, 2003, Japan [24]19BT-PABA88% showed reduced pancreatic exocrine function, none of whom reported steatorrhea.Impaired pancreatic exocrine function improved after CST in 3 of 6 patients.42% with DM.CST subsequently improved insulin secretion and glycaemic control in 3 of 5 patients.Ito, 2011,Japan [13]102BT-PABAPancreatic exocrine dysfunction was noted in 74.0% of all patients.Pre-existing DM-group A (n = 35, 34.3%). New onset DM-group B (n = 58, 56.8%)After steroid therapy (1.5 years)-DM group C (n = 9, 8.8%).Lee, 2018, South Korea [29]138-Not determined.45.7% had DM: 28.3% had pre-existing DM, and 17.4% had newly diagnosed DM (simultaneous onset or diagnosis during follow-up).Noguchi, 2020, Japan [26]61-Not determined.71% had DM. Anti-diabetic treatment became unnecessary in a quarter of patients with concurrent DM after 2 years of CST. DM was newly diagnosed in 12% of patients without DM at AIP diagnosis during CST.Masuda, 2014, Japan [27]31-Not determined.35% had DM. Six months after starting CST, DM was worsening in 9 of 11 DM patients.Kubota, 2018, Japan [30]97-Not determined.New-onset DM was noted in 26.2% of patients.Present study, 2022, Sweden73FE-1Prevalence of PEI at diagnosis: 72.7%. Prevalence of PEI at the last control: 63.5%The cumulative incidence of DM was 17.9%, with a prevalence of DM at diagnosis of 32.8%.BT-PABA = N-benzoyl-L-tyrosyl-p-aminobenzoic acid; FE-1 = fecal elastase-1; AIP = autoimmune pancreatitis; CST = corticosteroid treatment; DM = diabetes mellitus; ICDC = International Consensus Diagnostic Criteria; PEI = pancreatic exocrine insufficiency.


## 5. Conclusions

Prevalence of exocrine and endocrine pancreatic impairment was high at the time of diagnosis of AIP. During follow-up, patients were at risk of PEI and DM regardless of pharmacological treatment, and PEI and DM might not be reversed, as has been previously suggested. Besides OOI as a previously known risk factor for PEI, AIP patients with biliary stenting might be at a greater risk for PEI development at last contact. Obstructive jaundice is an independent risk factor for the development of DM at any time of the AIP disease course. Life-long follow-up of patients with AIP is advisable, especially for those with an initial presentation with obstructive jaundice and other organ involvement.

## Figures and Tables

**Figure 1 jcm-11-03724-f001:**
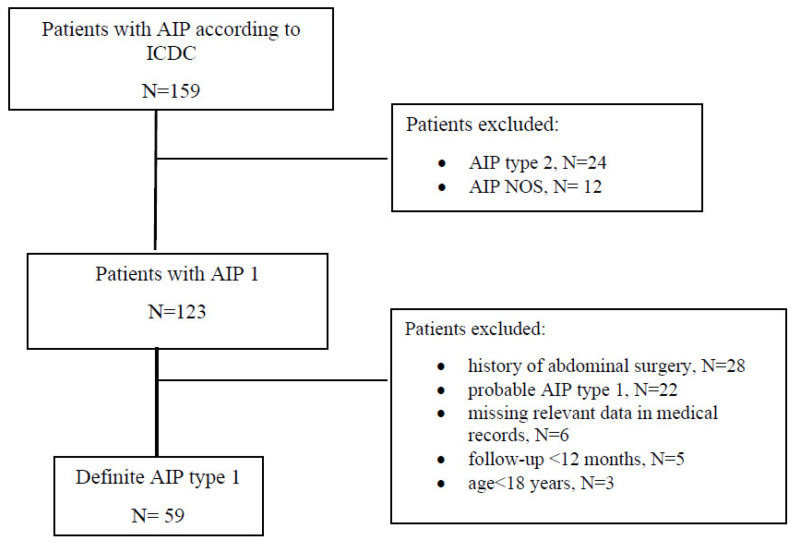
Flow chart of patients. AIP = autoimmune pancreatitis; ICDC = International Consensus Diagnostic Criteria; N = number of patients.

**Figure 2 jcm-11-03724-f002:**
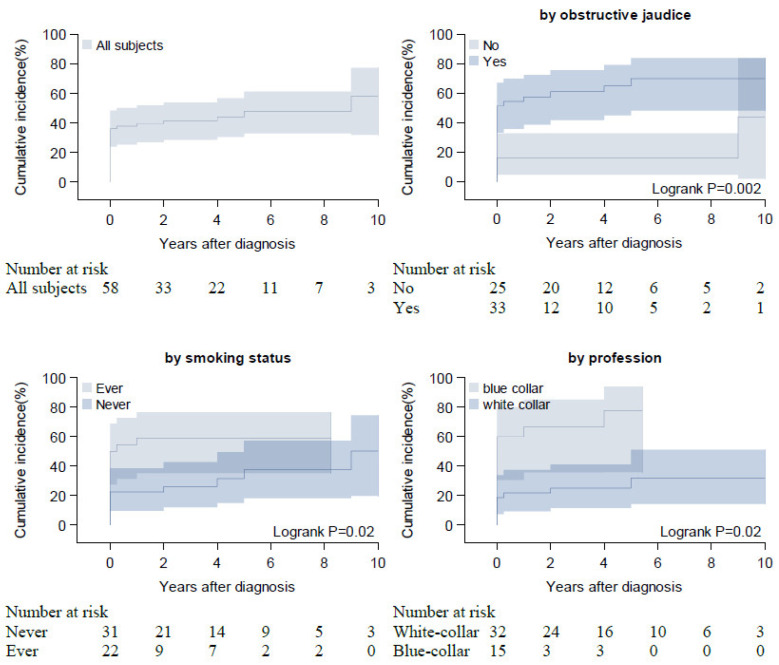
Prevalence of DM at diagnosis of AIP and cumulative incidence during the follow-up (stratified by obstructive jaundice, profession, and smoking status). Smoking status missing for 5 patients, profession for 11 patients; Analysis is restricted to 58 patients with available information about DM at diagnosis.

**Table 1 jcm-11-03724-t001:** Characteristics of patients with AIP type 1.

Patients, N = 59	Total (100%)	N (%)		Total (100%)	N (%)
Female, n (%)	59	19 (32.2)	**PEI, n (%)**		
Age at diagnosis (median, IQR)	59	65 (49–71)	at diagnosis	44	32 (72.7)
Follow-up (months)	59	62 (38–104)	FE 1 (median, IQR)		71.0 (15.0–223.8)
Alcohol consumption > 5 U	54	1 (1.9)	mild		9 (20.5)
Smoking	54		severe		23 (52.3)
Current		3 (5.6)	at last contact	52	33 (63.5)
Former		19 (35.2)	FE 1 (median, IQR)		101.50 (17.8–349.3)
Blue-collar profession	48	15 (31.3)	mild		7 (13.5)
AIP symptoms at diagnosis	59		severe		26 (50.0)
Abdominal pain		26 (44.1)	New PEI		3 (7.1)
Weight loss		20 (33.9)	PEI recovery		4 (9.5)
Acute pancreatitis		12 (20.3)	PEI class change (mild to severe)	42	4 (9.5)
Obstructive jaundice		33 (55.9)	PEI class change (severe to mild)	42	5 (11.9)
New onset diabetes	58	14 (23.7)	Diabetes mellitus, n (%)	58	
Incidental finding		7 (11.9)	at diagnosis		19 (32.8)
IgG4 serology	57		at last contact		27 (46.5)
positive		17 (29.8)	Clinical remission at last contact	55	51 (92.7)
2× normal value		13 (22.8)	Radiological remission at last contact	54	
Pancreatic enlargement on imaging	57		complete		43 (79.6)
diffuse		33 (57.9)	partial		6 (11.1)
focal		19 (33.3)	AIP treatment, n (%)	59	
OOI	59	56 (94.9)	No		8 (13.6)
Relapse	55	31 (56.4)	Steroid		49 (83.1)
1×		19 (32.2)	Azathioprine		9 (15.3)
2×		3 (5.1)	Rituximab		10 (16.9)
3×		5 (8.5)	Biologics		2 (3.4)
4×		3 (5.1)	Maintenance treatment	59	22 (37.3)
5×		1 (1.7)			
			Biliary stent, n (%)	59	27 (45.8)

AIP = autoimmune pancreatitis; PEI = pancreatic exocrine insufficiency; DM = diabetes mellitus; OOI = other organ involvement, U = units of alcohol.

**Table 2 jcm-11-03724-t002:** Factors associated with DM at univariate and multivariable analysis.

	Diabetes Mellitus at Diagnosis				Diabetes Mellitus at Diagnosis or during Follow-Up			
	Univariate		Multivariable *		Univariate		Multivariable *	
	OR (95% CI)	*p*-Value	OR (95% CI)	*p*-Value	HR (95% CI)	*p*-Value	HR (95% CI)	*p*-Value
BMI ≥ 25 kg/m^2^	4.46 (1.31–15.2)	0.017			2.36 (1.04–5.32)	0.039		
Blue-collar profession	6.50 (1.67–23.4)	0.007			3.35 (1.36–8.26)	0.009	2.38 (0.92–6.16)	0.074
Smoking	3.47 (1.02–11.8)	0.046	4.92 (1.22–19.9)	0.025	2.07 (0.90–4.75)	0.086	2.30 (0.94–5.63)	0.070
Weight loss	2.29 (0.73–7.16)	0.154			1.92 (0.90–4.10)	0.091		
Obstructive jaundice	6.90 (1.73–26.6)	0.006	9.47 (2.06–43.5)	0.004	3.87 (1.46–10.3)	0.007	4.92 (1.75–13.9)	0.003
PEI at diagnosis	6.60 (0.76–57.7)	0.088			4.25 (0.97–18.7)	0.056		
Diffuse	3.96 (1.10–14.2)	0.035			2.33 (0.97–5.58)	0.059		
Stent	4.33 (1.34–14.0)	0.014			3.50 (1.52–8.09)	0.003		

* Only variables with *p* < 0.10 are presented in the table and retained in the multivariable models; BMI = body mass index; DM = diabetes mellitus; PEI = pancreatic exocrine insufficiency, OR = odds ratio; HR = hazards ratio; diffuse = diffuse pancreatic enlargement.

## Data Availability

The data collected and analysed during the current study are available from the corresponding author on reasonable request.

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
