# Peer review of "Exocrine and Endocrine Insufficiency in Autoimmune Pancreatitis: A Matter of Treatment or Time?"

_jcm, 2022, doi:10.3390/jcm11133724_

Round 1
Reviewer 1 Report
In their study, Nikolic et al. have investigated prevalence and cumulative incidence of exocrine and endocrine pancreatic insufficiency in patients with autoimmune pancreatitis (AIP) type 1. The aim was to identify risk factors of both, paying particular attention to the effects of pharmacological treatment.
Therefore, the authors performed a retrospective analysis of altogether 59 patients with AIP type 1. They found that the prevalence of pancreatic exocrine insufficiency (PEI) and diabetes mellitus (DM) was high at diagnosis (72.7% and 34.6%, respectively). The cumulative incidence of DM was 18.4% (median follow-up: 62 months). In multivariate analysis, smoking and obstructive jaundice were identified as risk factors for DM both at diagnosis and during follow-up. The risk of PEI was increased in patients with other oragn involvements and with biliary stenting. There was no strong association between pharmacological treatment and occurrence of DM and PEI. The authors conclude that a life-long follow-up of patients with AIP is advisible, especially for those with an initial presentation with obstructive jaundice and other organ involvement.
This is an interesting and well conducted study. The findings are new and clinically important. The following few points should still be addressed:
1. Why did the authors exclude patients with AIP type 2? They could be analysed as a separate group.
2. Did the authors observe any cases of pancreatic cancer in their cohort during follow-up?
3. Please double-check all numbers in the supplemental tables. I've got confused by various minor differences. Examples: Table S2, first line: 58 patients, 18 of them with DM at diagnosis, +8 during follow-up. Main text; 59 patients; 3.2 (page 4): 14+6=20 patients with DM at diagnosis, +7 at follow-up.
4. Table 3 is not easy accessible and should be shortened.
Author Response
Reviewer 1:
In their study, Nikolic et al. have investigated prevalence and cumulative incidence of exocrine and endocrine pancreatic insufficiency in patients with autoimmune pancreatitis (AIP) type 1. The aim was to identify risk factors of both, paying particular attention to the effects of pharmacological treatment. Therefore, the authors performed a retrospective analysis of altogether 59 patients with AIP type 1. They found that the prevalence of pancreatic exocrine insufficiency (PEI) and diabetes mellitus (DM) was high at diagnosis (72.7% and 34.6%, respectively). The cumulative incidence of DM was 18.4% (median follow-up: 62 months). In multivariate analysis, smoking and obstructive jaundice were identified as risk factors for DM both at diagnosis and during follow-up. The risk of PEI was increased in patients with other organ involvements and with biliary stenting. There was no strong association between pharmacological treatment and occurrence of DM and PEI. The authors conclude that a life-long follow-up of patients with AIP is advisable, especially for those with an initial presentation with obstructive jaundice and other organ involvement.
This is an interesting and well conducted study. The findings are new and clinically important. The following few points should still be addressed:
Comment 1. Why did the authors exclude patients with AIP type 2? They could be analyzed as a separate group.
Authors’ answer: Thank you for comment. AIP type 2 is a different entity of AIP with lower rate of relapse, lower rates of PEI and DM and rare other organ involvement (however, commonly associated with inflammatory bowel disease). We hypothesized that AIP type 2 inclusion would underestimate the associations and introduce bias. We addressed AIP type 2 and calculated prevalence of PEI and DM at diagnosis and follow up in recently published article: PMID 35526270; Nikolic S, Lanzillotta M, Panic N, et al. Unraveling the relationship between autoimmune pancreatitis type 2 and inflammatory bowel disease: Results from two centers and systematic review of the literature. United European Gastroenterol J. 2022;10(5):496-506. doi:10.1002/ueg2.12237 https://pubmed.ncbi.nlm.nih.gov/35526270/
Comment 2. Did the authors observe any cases of pancreatic cancer in their cohort during follow-up?
Authors’ answer: Thank you for comment. In patients who were followed due to AIP, we observed no cases of pancreatic cancer. In patients who were presented at multidisciplinary team meeting due to suspicious mass, which after surgery and histological analysis resulted in pancreatic cancer and AIP, we observed two cases and addressed this in our article: PMID: 34915509, Nikolic S, Ghorbani P, Pozzi Mucelli R, et al. Surgery in Autoimmune Pancreatitis. Dig Surg. 2022;39(1):32-41. doi:10.1159/000521490 https://pubmed.ncbi.nlm.nih.gov/34915509/
Comment 3. Please double-check all numbers in the supplemental tables. I've got confused by various minor differences. Examples: Table S2, first line: 58 patients, 18 of them with DM at diagnosis, +8 during follow-up. Main text; 59 patients; 3.2 (page 4): 14+6=20 patients with DM at diagnosis, +7 at follow-up.
Authors’ answer: We thank this reviewer for pointing the discrepancy. We controlled the whole dataset and found a coding error responsible for this discrepancy. We repeated the whole analysis and found very similar results. Though, at multivariable analysis, adjustment for smoking did not eliminate the association between blue-collar professions and DM. We corrected all the tables and figures and added a new cumulative incidence plot of DM by profession in figure 2.
Comment 4. Table 3 is not easily accessible and should be shortened.
Authors’ answer: Thank you for your comment, we agree. We made changes (shortening).
Author Response
Reviewer 2:
The paper focused on PEI and DM prevalence in AIP before and after treatment. The study showed that PEI and DM prevalence did not change significantly in tested cohort. I have some important questions.
Comment 1: The diagnosis of PEI was based on the value of FE-1. The authors categorized the level of FE-1 as mild and severe PEI. What is the basis for such categorization?
Authors’ answer: Thank you for question. Study which inspired us to design our analysis on this topic was by Frulloni et al (reference number 6). We encountered in their methods above mentioned classification to mild and severe PEI, which we reproduced. We add the reference. Differentiation between mild and severe is widely accepted nowadays, especially after publication of United European Gastroenterology evidence based guidelines for the diagnosis and therapy of chronic pancreatitis (HaPanEU) in 2017.
Comment 2: Why don't the authors present the elastase-1 values in Table 1?
Authors’ answer: Thank you for question. We felt it would overcrowd the data. We added the median values at diagnosis and last contact upon your suggestion.
Comment 3: According to International Consensus Diagnostic Criteria for Autoimmune Pancreatitis Guidelines of the International Association of Pancreatology, Level1 criteria for Type a AIP include 5 points, among them the level of IgG4. Meanwhile, we see that IgG4 positivity does not occur in all patients (negative in 45.5% of patients).
Authors’ answer: Thank you for comment. We agree. IgG4 serum level alone lacks sensitivity and specificity, but can be helpful to establish the diagnosis, and therefore should be measured if IgG4-related gastrointestinal disease is suspected. To diagnose IgG4-related disease, current recommendations propose a comprehensive workup, including histology, organ morphology at imaging, serology, search for other organ involvement, and response to glucocorticoid treatment. IgG4 serum levels seem to have diagnostic value when the level is higher than four times the upper level of normal, which is the case in only a minority of patients. We cited recently published European Guideline on IgG4-related digestive disease – UEG evidence-based recommendations (lead by members of our team).
Comment 4: Why the authors do not present the results of Fisher's test for the total number of PEI and DM? (first line in Table 1 and 2)? The p-value would be the response to the treatment effect on the incidence of PEI and DM.
Authors’ answer: Thank you for comment. We refrained from comparing patients receiving the treatment and those without due to possible bias by treatment indication. We have though performed Fisher’s test comparing different treatment options as well as maintenance treatment with prevalence of PEI, supplementary tables 1 and 2.
Comment 5: Authors point to obstructive jaundice is an independent risk factor for the development of DM at any time of the AIP disease course. a/ How was jaundice diagnosed? b/ An attempt to clarify the link would be appreciated. c/ It would be interesting to compare the incidence of cholestasis with the bilirubin concentration.
Authors’ answer: Thank you for your comment. a/ in our routine clinical practice jaundice is diagnosed clinically (inspection of sclerae and skin) and confirmed by higher bilirubin concentration and we used the data from medical charts. b/ Obstructive jaundice in AIP could be a result of inflammation (periductal, storiform fibrosis, IgG4 positive cells, obliterative phlebitis) and/or via compression of inflammatory swelling on pancreatic part of biliary duct. IgG4 related cholangitis is a common OOI and inflammatory stenosis on biliary duct could have partly mediated the obstructive jaundice. c/we absolutely agree that that would be an interesting study, maybe revealing bilirubin as a useful marker of disease intensity in future.
Comment 6: What is the alcohol consumption measure expressed as U?
Authors’ answer: Thank you for question. That is a unit of alcohol which equals to 10 ml of pure alcohol in a beverage. We explain to patients that 330 ml of lager beer is 1 U, 250 ml of wine is around 2.5 U, and a shot is 1 U.
Comment 7: What clinical (invasive, non-invasive) tests have been performed to confirm the diagnosis (parenchymal and ductal structure)?
Authors’ answer: Thank you for question. Diagnosis of AIP was made according to ICDC and European criteria (references 1 and 2).
We routinely measure IgG4 despite of the fact that IgG4 serum level alone lacks sensitivity and specificity, but is helpful to establish the diagnosis, and therefore is routinely measured in all patients in whom IgG4-related gastrointestinal disease is suspected.
Parenchymal changes suggestive of AIP and routinely used in our center are: (i) Diffuse or (multi-) focal enlargement with loss of the normal multilobulated pattern (‘sausage-like’ shape); with diffuse involvement more frequent in type 1 (compared to more frequent focal involvement in AIP type 2). (ii) Altered imaging characteristics, such as lower signal intensity (SI)/echogenicity on unenhanced T1-w MRI/(E)US, respectively, moderately higher SI on T2-w MRI, impeded diffusion on MRI, and increased 18F-fluorodeoxyglucose (FDG)-uptake on PET-CT compared with normal parenchyma. (iii) Rectangular shape of the tail (‘cut-tail sign’). (iv) Thin peripancreatic edematous rim or progressively enhancing true capsule. Ductal changes suggestive of AIP and routinely used in our center are: (i) Long-segment or multifocal main pancreatic duct (MPD) involvement (narrowing or vanishing) without upstream dilatation or other signs of obstructive pancreatitis. (ii) Skip lesions, i.e. ≥2 involved MPD-segments separated by a normal MPD-segment.
Since all these details are described in references 1 and 2 we decided not to explain (and repeat) all details because it may influence readability and fluency of the manuscript and divert reader’s attention.